# Omicron Variant Generates a Higher and More Sustained Viral Load in Nasopharynx and Saliva Than the Delta Variant of SARS-CoV-2

**DOI:** 10.3390/v14112420

**Published:** 2022-10-31

**Authors:** Beathe K. Granerud, Thor Ueland, Andreas Lind, Arne Søraas, Børre Fevang, Anne Katrine Steffensen, Huda Al-Baldawi, Fridtjof Lund-Johansen, Pål Aukrust, Bente Halvorsen, Tuva B. Dahl, Susanne Dudman, Fredrik Müller, Jan Cato Holter

**Affiliations:** 1Institute of Clinical Medicine, University of Oslo, 0316 Oslo, Norway; 2Department of Microbiology, Oslo University Hospital, 0424 Oslo, Norway; 3Research Institute of Internal Medicine, Oslo University Hospital, 0424 Oslo, Norway; 4K.G. Jebsen Thrombosis Research and Expertise Center, Faculty of Health Sciences, University of Tromsø, 6050 Tromsø, Norway; 5Section of Clinical Immunology and Infectious Diseases, Oslo University Hospital, 0424 Oslo, Norway; 6Department of Immunology, Oslo University Hospital, 0424 Oslo, Norway; 7ImmunoLingo Convergence Centre, University of Oslo, 0316 Oslo, Norway; 8Division of Critical Care and Emergencies, Oslo University Hospital, 0424 Oslo, Norway

**Keywords:** viral load, Omicron variant, Delta variant, Norway, SARS-CoV-2

## Abstract

The Omicron variant of SARS-CoV-2 spreads more easily than earlier variants, possibly as a result of a higher viral load in the upper respiratory tract and oral cavity. Hence, we investigated whether the Omicron variant generates a higher viral load than that of the Delta variant in saliva and nasopharynx. Both specimens were collected from 52 Omicron and 17 Delta cases at two time points one week apart and analyzed by qRT-PCR. Viral load was measured as 10 log RNA genome copies per 1000 human cells according to the WHO reference standard. We found that Omicron cases carried a higher viral load and had more sustained viral shedding compared to the Delta cases, especially in the nasopharynx.

## 1. Introduction

The SARS-CoV-2 Omicron variant (B1.1.529) was first reported in South Africa on 24 November 2021 [1]. Just one week later, on 30 November 2021, a laboratory in Oslo suspected and later confirmed Norway’s first case of the SARS-CoV-2 Omicron variant. The patient was present at a Christmas party where one of the other participants had recently returned from a trip to South Africa. The result was one of the first documented outbreaks of the SARS-CoV-2 Omicron variant outside of Botswana and South Africa [2]. The Norwegian Institute of Public Health (NIPH) reported an attack rate of 74 %, among which 96% of affected individuals were fully vaccinated, possibly indicating that the Omicron variant was more transmissible than the Delta variant [2]. Other factors that can explain the high attack rate included reduced susceptibility to neutralizing antibodies [3]; environmental factors, such as prolonged indoor exposure [4]; or a high rate of asymptomatic carriage [5].

In response to this outbreak, nasopharyngeal swabs (NPS), saliva and blood were prospectively collected from 75 individuals, of whom 52 were confirmed to be infected with the Omicron variant, 17 with the Delta variant and 6 negative according to PCR. Findings indicated that the increased transmissibility observed in association with the Omicron variant was not due to evasion from vaccine-induced immunity [6]. Therefore additional studies on the kinetics of infectious viral load are needed to better understand the mechanisms behind the distinct transmissibility of SARS-CoV-2 variants, as well as the effect of vaccination, to inform public health guidance and the choice of optimum specimens for diagnostic testing. 

Although recent meta-analyses published during the pandemic support the use of NPS over saliva [7,8], only a few studies have examined viral load in saliva and NPS in Delta vs. Omicron cases. Importantly, these qRT-PCR (quantitative reverse-transcriptase real-time PCR) studies are hampered by the use of Ct values as a proxy for viral load [9,10,11,12], the use of an in-house standard with no traceability [13,14] or the lack of normalization of SARS-CoV-2 RNA against human DNA [10,11,12]. 

Hence, in this study, we compared the viral load in saliva and NPS from 52 Omicron and 18 Delta cases collected at two time points one week apart; viral load was measured by qRT-PCR using the WHO international reference standard for SARS-CoV-2 dilution series method. 

## 2. Materials and Methods

A detailed description of the cohort and sampling procedures were previously published in [6]. In brief, we collected NPS and drool-spit saliva at the patient’s home or at an outpatient clinic at Oslo University Hospital after a median of 7 days (range 3-10 days) and 14 days (range 9–18 days) after symptom onset in individuals infected with the Omicron or Delta variant. All of the patients were outpatients living in Oslo or the surrounding county, Viken. None received treatment.

Informed consent was obtained from all individuals. The study was approved by the Regional Committee for Medical and Health Research Ethics in South-Eastern Norway (reference numbers 124170 and 106624).

### 2.1. Viral Load Analysis

We inactivated, extracted and analyzed all samples as previously described [6]. Owing to their viscosity, saliva samples were diluted 1:2 in lysis buffer (Qiagen, Hilden, Germany) containing N-acetylcystein (10 g/L) before extraction. Some samples required extra dilution with PBS (phosphate-buffered saline) because automated extraction instruments misinterpreted increased turbidimetry as “too much volume”. 

In brief, we performed real-time PCR using E-gene for SARS-CoV-2 RNA quantification [6] and HPRT1 gene (CELL Control r-gene, ref 71–106, BioMerieux, France) for human DNA quantification, using the first WHO International Standard for SARS-CoV-2 (reference standard 20/146, NIBSC, UK) and the internal kit standards for relative quantification. Standards were included to each plate, and PCR efficiency and interplate variation were assessed using Ct-values for standards, slope and Y-intercept. Finally, we determined the viral load as log 10 virus RNA copies per 1000 human cells. 

### 2.2. Statistical Analysis

A linear correlation mixed model was used to assess the association between viral load (log 10-transformed), symptom days, virus variant and sample material. Only Hb and lymphocyte counts showed a normal distribution of the biochemical variables and are presented as mean ± SD and compared with the Student’s t-test results also performed for age. The rest of the biochemical data were skewed and are presented as median (25th and 75th percentile) and compared with Mann–Whitney U-test results. A chi-square test was used to analyze differences in categorical demographics. Differences in age were compared using Student’s *t*-test. *p*-values were two-sided and considered significant when <0.05.

## 3. Results

### 3.1. Cohort Characteristics

Delta and Omicron cases were comparable in terms of demographic, clinical and laboratory infection parameters (Table 1). All but one patient in each group had received at least one dose of an mRNA COVID-19 vaccine. We received both sample materials from all patients at inclusion, except one saliva sample in the Delta group. One week after inclusion, several patients opted out of NP sampling or saliva sampling (see Table 1).

### 3.2. Viral Load in Nasopharyngeal Swabs and Saliva

Viral load relative to days after symptom onset, virus variant and specimen is shown in Figure 1. We previously reported data on viral load in NPS (Figure 1A,B) [6], and herein, we report comparative data on viral load in saliva (Figure 1C,D). 

A negative and similar correlation was observed for viral load and symptom days in NPS (Omicron r = −0.23, *p* < 0.001; Delta r = −0.23, *p* < 0.001) and in saliva (Omicron r = −0.28, *p* < 0.001; Delta r = −0.14, *p* = 0.34), except for the Delta variant in saliva, for which the correlation was slightly weaker (Figure 1C). Importantly, a higher viral load and a prolonged duration of SARS-CoV-2 shedding in the upper respiratory tract, as judged by saliva and NPS in our study, was observed for the Omicron variant compared to the Delta variant, especially in NPS (*p* < 0.05).

## 4. Discussion

Since the onset of the COVID-19 pandemic in December 2019, five circulating VOCs (variants of concern) of SARS-CoV-2 have been identified, all differing in terms of transmissibility, disease severity, immune escape and diagnostic escape due to mutations in the spike gene [15]. The newest VOC, the Omicron variant, spreads more easily than the other variants, including the Delta variant [16].

In the present study, our findings, based on WHO´s standard reference method for qRT-PCR quantification of SARS-CoV-2 in respiratory materials, indicate that a higher and more sustained viral load may serve as possible explanation for this increased transmissibility of the Omicron compared to the Delta variant. This contrasts the results of other studies that reported a lower or similar viral load of the Omicron variant in NPS [17,18,19,20,21] and saliva [17] compared to the Delta variant. None of these studies normalized were against human DNA or considered days since symptom onset. However, one study reported a higher amount of SARS-CoV-2 in exhaled air surrounding Omicron patients than that surrounding Delta patients, although there was no difference in the viral load in NPS [21]. However, as estimation of viral load is dependent on a number of factors, such as PCR sensitivity, linearity/efficiency, sampling techniques, pretreatment (e.g., different methods for dissolving mucus) and extraction methods [22,23], a direct comparison of results between studies using different protocols is challenging. In our study, normalization against a human gene and the use of an internationally validated standard strengthens our conclusion in comparison to those reported in previous studies [24]. Another possible explanation for varying results is that the time from exposure to symptom onset and sampling is unknown. However, in our study, it was likely that all Omicron cases, except household contacts later included in the Omicron group, were exposed on the same day to one person who had just returned from South Africa [2], resulting in a high level of integrity of these time-dependent variables. However, time from exposure to symptom onset and sampling for Delta cases was more uncertain. Despite the use of a linear correlation mixed model to compensate for differences in symptom duration, this method cannot compensate for differences in replication kinetics. If the Delta variant had an undisputedly higher initial replication rate and steeper viral load decay than the Omicron variant, this would have affected our results. However, as the Omicron variant possibly has a higher [25,26] or similar [27,28] initial replication rate and viral load decay relative to the Delta variant, this is not the case.

Furthermore, as the viral load for both variants is higher in NPS than in saliva and remains so for a long period, our findings further support the suggestion that NPS is preferable to saliva [7,8] as an optimal specimen. 

The impact of viral load, as measured by qRT-PCR, on infectiousness varies considerably among individuals [29]. A high viral load (i.e., low Ct value) increases the probability of transmission—especially in household contacts—but is also dependent on the SARS-CoV-2 variant, the patient’s age and other individual factors [29,30]. Daily sampling of NPS and saliva from household contacts and confirmed cases from the time of exposure to different variants of the virus, combined with the use of the WHO protocol for estimation of viral load, could have led to more consistent data in this field.

## Figures and Tables

**Figure 1 viruses-14-02420-f001:**
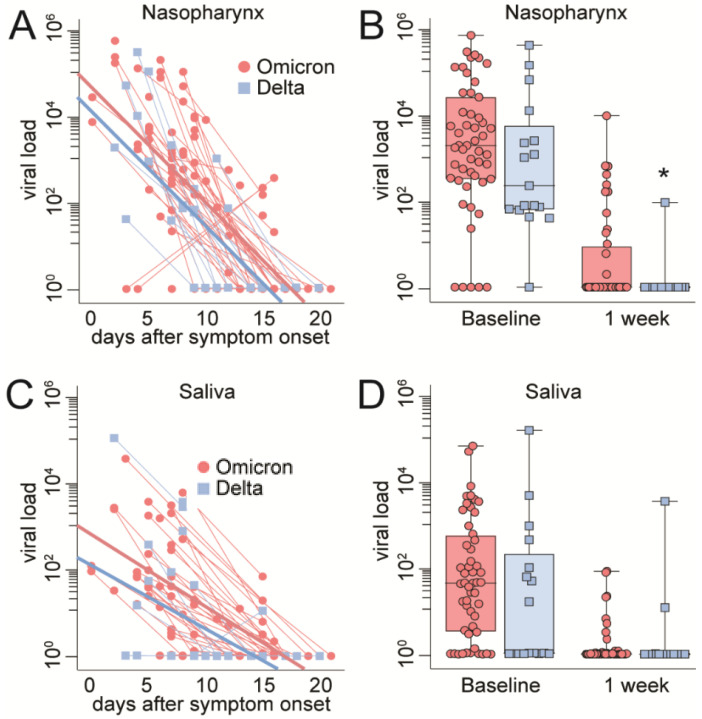
Viral load in nasopharyngeal swabs vs. saliva. (**A**,**C**): Correlation between viral load and days after symptom onset in NPS or saliva. Thick red/blue lines represent regression curves for the whole group (Delta or Omicron), whereas thin red/blue lines represent paired samples. (**B**,**D**): Viral load in NPS or saliva, shown as Tukey plots at inclusion (baseline) and one-week follow-up. Figure 1A,B was previously published in [6]. * *p* < 0.05 Omicron vs. Delta was adjusted for symptom duration.

**Table 1 viruses-14-02420-t001:** Demographic, clinical and laboratory findings at inclusion.

	Omicron, n = 52	Delta, n = 17	*p*-Value
Age, years	38.9 ± 12.3	39.8 ± 8.2	0.77
Male gender	25 (48.1%)	6 (35.3%)	0.36
Saliva (inclusion/1 week)	52 (100%)/45 (86.5%)	16 (94.1%)/14 (82.4%)	0.26/0.70
NP (inclusion/1 week)	52 (100%)/46 (88.5%)	17 (100%)/14 (82.4%)	0.25/1.00
Vaccinations, (n = 0/1/≥2)	1/3/48	0/0/17	0.19
Symptom days, inclusion	6 ± 3	7 ± 3	0.67
Symptom days, 1 week	15 ± 3	13 ± 4	0.23
Hgb (g/dL)	14.4 ± 1.1	13.9 ± 1.2	0.19
WBC (×10^9^/L)	5.4 ± 1.5	5.7 ± 1.6	0.58
Neutrophils (×10^9^/L)	2.9 ± 1.1	3.1 ± 1.1	0.25
Lymphocytes (×10^9^/L)	2 ± 0.6	1.9 ± 0.6	0.74
CRP (mg/L)	1.8 (0.7, 4.9)	1.9 (1, 4.4)	0.95
Ferritin (µg/L)	119 (66, 221)	159 (72, 201)	0.87
eGFR	110 ± 13	106 ± 10	0.13

Data are presented as n (%), mean ± SD or median (25–75th percentile) unless otherwise indicated. Abbreviations: Hgb, hemoglobin; WBC, white blood cells; CRP, C-reactive protein; eGFR, estimated glomerular filtration rate. Included samples are noted as numbers at inclusion (% total patients)/numbers after 1 week (% total patients).

## Data Availability

Not applicable.

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
