# Peer review of "Omicron Variant Generates a Higher and More Sustained Viral Load in Nasopharynx and Saliva Than the Delta Variant of SARS-CoV-2"

_viruses, 2022, doi:10.3390/v14112420_

Round 1
Reviewer 1 Report
I appreciate for entrusting me with the review of this article.
This paper shows that the Omicron strain has a higher virus burden in the nasopharynx than the Delta strain. I would like to make a few comments on the study.
Line 84-89:
Some of the statistics mentioned here are not shown in the results.
I don't think the statistics for symptom days, viral variants, or sample material are mentioned in the Result section. Also, the analysis results regarding the categorical variables and patient age are not listed in the Result. The results of these statistical tests should be clearly shown if the authors write about them.
Line 110-112:
It is stated that a higher viral load was observed in the Omicron strain, but there is no statistical evidence for this, so please add it. it should also be noted whether there was any difference in viral load between Omicron and Delta in saliva samples.
Line 135.
As the author elaborates, the effect of treatment may affect viral load. It would be useful to note whether the patients in this study were treated or untreated at the time of the initial collection and the follow-up.
Table 1
Regarding Saliva (n=inclusion/1week) and NP (n=inclusion/1 week), what does /1 week indicate here? Does it mean the percentage of inclusion within one week from the start of recruitment? Please clarify this.
Do samples 1 and 2 refer to the initial sample and the follow-up sample? Please clarify them.
CRP and Ferritin appear to be shown as median and quartiles, but this is not explained. Also, the parentheses are not aligned with [ on the left and ) on the right, please correct this.
Author Response
Thank you for taking your time to give us a good and detailed review. We apologize for the poor description in the statistical analysis section and have redone this section in the revision. We now include the distribution of the continuous variables (i.e they were skewed or not), the choice of summary statistics and comparison tests (i.e mean +/- SD and Students t-test for normally distributed data and median [25th, 75th percentile] and Mann-Whitney U-test for skewed in table 1. We are also sorry that some of the p-values had fallen out of the manuscript, these are now included. Regarding your other comments:
We have included and clearified the information you were asking for, and corrected the typos in table 1 where the parentheses were not aligned.
Author Response
Thank you for your time and for giving us a good and detailed review. We are sorry for the confusion regarding how many samples were used in the serology study and how many were used in the PCR study. We have rewritten the paragraph in the reviewed document, hopefully giving the readers a better understanding of the process. Unfortunately we can not add results related to the control group to the manuscript. The reason for this is that several of the household contacts got infected by their familiy, and are therefor included in the Omicron group. The rest were PCR negative in all samples and are therefore not interesting in this context. Regarding your question about the number of samples from patients infected with the Delta variant: There were initially 18 patients. Samples from 14 were used in the serology study (the rest were outliers) and samples from 18 were used in this study. Of the 18, one turned out to be PCR negative in all samples and was therefore excluded from this study.
Reviewer 3 Report
Granerud et al. reported the viral load discrepancy between omicron variants and the delta variants of SARS-CoV-2 in nasopharynx and salina of infected patients. The question they raised is pretty attractive, and the enrolled study cohort is quite straightforward to investigate the topic. For data analysis, they enrolled the human gene HPRT1 as an internal control to normalize the viral load and they also used WHO reference standards to adjust their readout, which makes the data more convincing. This is a useful study and the conclusion is instructive.
General comments:
1. The integrity of the introduction. In lines of 35-37, “96 % of affected individuals were fully vaccinated, possibly meaning that the Omicron variant was more transmissible than the Delta variant”. Here, the authors should also clarify other possible reasons, i. e. the omicron may have reduced susceptibility to neutralizing antibodies induced by SARS-CoV-2 infection or vaccination (Lin-Lei Chen, Lu Lu et al., Clinical Infectious Diseases, Volume 74, Issue 9, 1 May 2022, Pages 1623–1630).
2. Over statement of findings. For the explanation of Figure 1, in the lines of 110-111, the conclusion of “sustained virus excretion” should be carefully addressed since the virus excretion is related to the virus copy in the NPS and that in the saliva, but also related to the total viral load in the patient. Here, the provided results can only support a higher viral load but not the “sustained virus excretion”.
3. The viral load of these two variants of SARS-Cov-2 may peak at different time points or the viral load kinetics may be slightly different after exposure to the symptom onset. It would likely result in a discrepancy in viral copy at sequential time points, which should be taken into consideration when making a comparison.
4. For the Materials and Methods, in lines 80-81, what’s the information (copy number? Source? …) for standards 1 and 2? Please clarify.
Author Response
Thank your for your time and for giving us a good and detailed review. Based on your comments we have included other possible reasons for a higher transmission rate. We are also sorry for overstating our findings and have rewritten this paragraph in the revised manuscript. Regarding your comment on viral load at different time points, we have partly adjusted for this by our choice of statistical method. But, we agree that the statistical method does not fully adjust for differences in virus kinetics. Hence, we have included this in the discussion. We have also included information about the source of the HPRT1-kits standards.
Round 2
Reviewer 2 Report
I would like to thank authors for the effort they did.